# LAYER-WISE ANALYSIS IN EXPLORING THE NORMALIZATION STRATEGIES IN MAMBA

## ABSTRACT

The Mamba architecture achieves linear time and memory complexity in long-sequence modeling and vision tasks through a dynamic, input-conditioned state transition mechanism and hardware-efficient scan operations. However, as network depth increases, the state space model (SSM) component tends to amplify activation magnitudes during the forward pass, often leading to gradient explosion. To relieve this, we analyze training stability by tracking (i) the spectral norm of the output projection weights and (ii) the largest eigenvalue of the joint input-output covariance matrix, demonstrating the effectiveness of post-SSM in suppressing activation and gradient scale inflation. From the perspective of optimization efficiency, we use K-FAC to approximate the Fisher Information Matrix and show that pre-SSM significantly reduces the condition number of per-layer gradients, thereby accelerating convergence. Furthermore, we propose a composite normalization strategy (BN→SSM→LN), combining BatchNorm before input projection layer and LayerNorm after the SSM layer. We evaluate this strategy across a broad range of benchmarks. Experimental results demonstrate that the composite scheme consistently outperforms single or no normalization in both convergence speed and final accuracy. We hope this work provides both theoretical insights and empirical guidance for normalization in designing SSM-based models.

## 1 INTRODUCTION

Mamba Gu & Dao (2023) has attracted significant attention across a broad range of applications, including long-sequence modeling tasks such as speech and audio processing Ren et al. (2025), as well as domains like natural language processing (NLP) and computer vision (CV) (Liu et al., 2024), due to its strong capabilities in capturing long-range dependencies and computational efficiency. However, Mamba faces notable training challenges, as instability during training often causes the iteration to diverge, particularly as model parameters scale up (Dao & Gu, 2024). This phenomenon is illustrated in Figure 1, which shows that as the depth increases from 4 to 32 layers, the training process becomes more unstable, with gradient explosion and divergence occurring earlier in deeper networks.

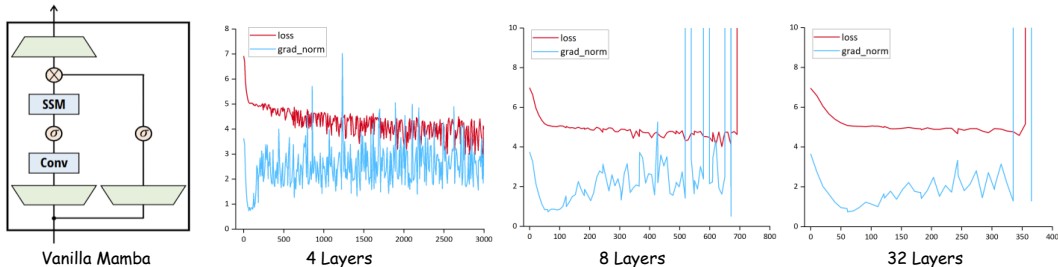

Figure 1: Training instability of Vanilla Mamba in Imagenet dataset. Red line indicates loss and blue line indicates gradient norm. As model depth increases from 4 to 32 layers, the training process becomes more unstable.

To better understand the source of instability, we examine the behavior of the SSM block. Specifically, we track the range of its input $\mathbf{x}$ and output $\mathbf{y}$ activations during training. As shown in Figure 2, two consistent phenomena emerge across both WikiText-103 and ImageNet: (1) the SSM amplifies input activations, and (2) with increasing steps, deeper layers first exhibit extremely large values, which soon explode to infinity after several iterations. These observations suggest that the intrinsic amplification effect of the SSM, together with the discrepancy in activation magnitudes between shallow and deep layers, is a key factor underlying training instability. Such amplification further indicates that the Mamba architecture inherently suffers from poor scale-invariance, which explains why deeper layers are more prone to instability.

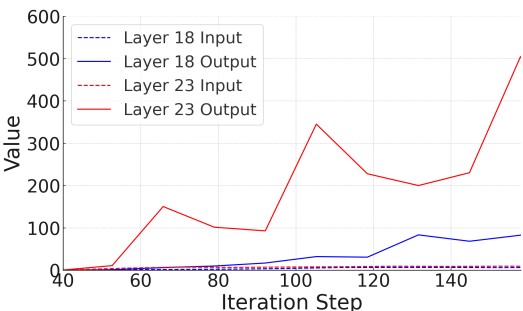

Figure 2: Input and Output Ranges of the SSM Module for vanilla mamba on the WikiText-103 Dataset.

Given that poor scale-invariance thus emerges as a fundamental limitation of Mamba, which directly contributes to the observed training instability. The success of deep neural networks has relied heavily on advances in training techniques, among which normalization of internal representations plays a central role (Hinton & Salakhutdinov, 2006; Nair & Hinton, 2010; Kingma & Ba, 2015). Normalization is widely recognized to stabilize and accelerate training by promoting scale-invariance, improving conditioning of the optimization landscape, and introducing beneficial stochasticity (Huang, 2022). We therefore turn to normalization as a potential remedy for the instability of Mamba.

Despite recent efforts Gu & Dao (2023); Ma et al. (2024); Liu et al. (2024) introducing different normalization layers into the Mamba architecture—such as Layer Normalization (LN) (Ba, 2016), Group Normalization (GN) (Wu & He, 2018), and Root Mean Square Normalization (RMSN) (Zhang & Sennrich, 2019), these adaptations have largely been task-specific, aiming to improve performance on individual benchmarks. However, there remains a lack of systematic analysis on the role of normalization in Mamba, particularly with respect to training dynamics. Beyond stability, another critical aspect of training dynamics is optimization efficiency, i.e., how fast and effectively the model converges. It is still unclear how different normalization choices, and their placements within the architecture, affect both the stability of training and the efficiency of optimization, leaving open the question of how to design principled normalization strategies for Mamba.

To bridge this gap, in this paper we focus on the two most common normalization positions in the Mamba architecture: after the SSM layer (Norm2) and before the input projection layer (Norm1), as illustrated in Figure 4. Building on our training dynamics analysis, we propose a two-stage hybrid normalization strategy: (1) Stage 1: Given that the SSM module amplifies activations and exacerbates instability, placing LN at Norm2 effectively stabilizes training and ensures convergence. Since different normalization methods exhibit complementary effects; for example, BN is known to improve optimization efficiency better than LN. (2) Stage 2: We further introduce BN at Norm1. This enhances optimization efficiency, enabling the model to reach higher accuracy faster.

Next, we conduct a layer-wise analysis to study the role of normalization in stabilizing and accelerating Mamba training. Specifically, we first examine the effect of applying LN at Norm2, by assessing two key indicators: (i) the spectral norm of the output projection weights and (ii) the maximum singular value of both the layer input covariance matrix and the layer output-gradient covariance matrix. These metrics reflect how LN at Norm2 enforces scale invariance (Ba, 2016) across layers and mitigates instability. After establishing training stability, we then analyze the effect of applying BN at Norm1 from an optimization perspective. To this end, we track the maximum singular value and condition number of the Kronecker-Factored Approximate Curvature (K-FAC) matrix (Huang

| Batch Norm | Group Norm | Instance Norm | Layer Norm | RMSN |
|---|---|---|---|---|
| 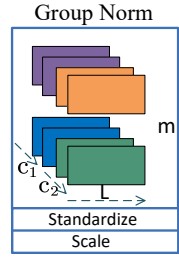 | 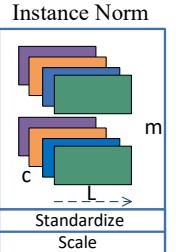 | 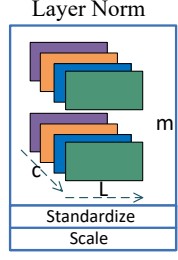 | 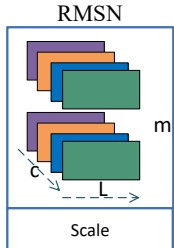 | |

Figure 3: **Normalization methods for sequence data** Each subfigure displays the dimensional information of a feature map, where mrepresents the batch axis, d the channel axis, L the sequence length axis. The dashed arrows indicate that the mean and variance are computed by aggregating the values across these dimensions.

et al., 2020), which approximates the Fisher Information Matrix (FIM). This analysis demonstrates that BN at Norm1 substantially improves optimization conditioning and accelerates convergence.

Building upon these insights, we propose a composite normalization strategy BN→SSM→LN. We evaluate this design across a diverse set of benchmark tasks, including image classification, object detection, semantic segmentation, long-sequence modeling, and natural language processing. The results show that our BN→SSM→LN configuration consistently outperforms baselines that use either a single normalization method or none at all.

To sum up, our contributions are summarized as follows:

- We demonstrate that applying LN at Norm2 plays a critical role in stabilizing training by suppressing activation and gradient scale explosion, as evidenced by tracking spectral norms and singular values of covariance matrices.
- We show that applying BN at Norm1 substantially improves optimization efficiency by reducing the condition number of the approximated Fisher Information Matrix (K-FAC), thereby accelerating convergence.
- We propose a composite normalization strategy BN→SSM→LN based on these insights. The design is derived from a general training dynamics perspective and theoretical analysis, which together establish a principled normalization guideline that combines the stability of LN and the efficiency of BN, yielding consistent improvements across diverse tasks.

## 2 RELATED WORK

### 2.1 LINEAR STATE SPACE MODELS

Transformers with quadratic-time attention (Vaswani et al., 2017) achieve strong performance but suffer from $O(n^2)$ complexity, which limits scalability in long-context applications. To address this, researchers have developed linear attention mechanisms (Choromanski et al., 2020; Katharopoulos et al., 2020) and state space models (SSMs) (Gu et al., 2021), both enabling efficient long-sequence modeling. Building on this line, Mamba (Gu & Dao, 2023) introduces a selective mechanism for content-aware state transitions and has inspired numerous extensions (Phung et al., 2024; Chiang et al., 2024; Wu et al., 2024; Pierro & Abreu, 2024; Zeng et al., 2024; Wei et al., 2024). However, these efficiency gains come with a major drawback: training instability. Unlike softmax attention, which inherently normalizes activations and gradients, Mamba's SSM block amplifies activations, violates scale-invariance, and often causes gradient explosion and divergence, particularly in deeper networks (Dao & Gu, 2024).

### 2.2 THE ROLE OF NORMALIZATIONS

The success of deep neural networks (DNNs) has relied heavily on normalization techniques that regulate the distribution of activations (Kingma & Ba, 2015; Ioffe & Szegedy, 2015). As illustrated in Figure 3, normalization methods differ in how they compute statistics across batch, channel, or feature

dimensions. For example, Batch Normalization (BN) standardizes activations across both batch and feature dimensions, effectively mitigating internal covariate shift and improving optimization efficiency (Ioffe & Szegedy, 2015; Wang et al., 2022). Layer Normalization (LN), in contrast, normalizes along the feature dimension within each sample, stabilizing hidden-state dynamics and preventing scale explosion across layers (Ba, 2016). While these methods have been extensively used in RNNs and Transformer-based architectures (Xiong et al., 2020; Shleifer et al., 2022; Han et al., 2021), their role in Mamba remains unclear. Several recent studies have attempted to insert normalization layers such as LN, GN, or RMSNorm into Mamba (Gu & Dao, 2023; Ma et al., 2024; Liu et al., 2024), but these adaptations are largely task-specific and lack a systematic analysis of training dynamics.

## 3 METHOD

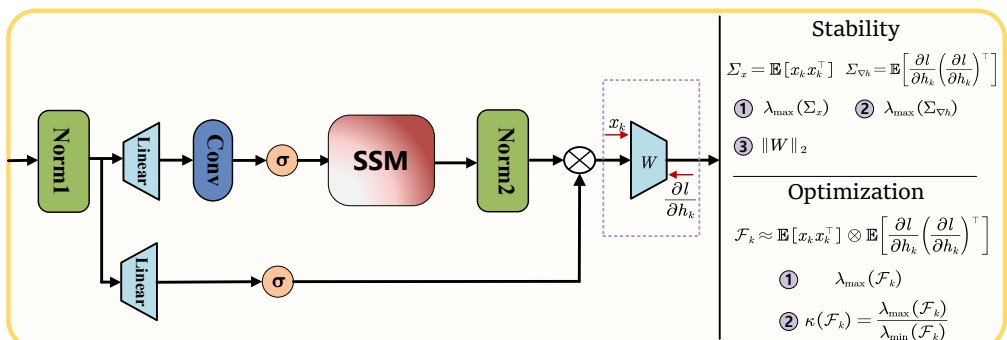

Figure 4: Mamba Normalization Framework. We conduct a layer-wise analysis of training stability and optimization in the Mamba architecture by examining the input and output-gradient of the output projection layer.

To systematically investigate the role of normalization in Mamba, we focus on the two most common insertion points: **Norm1**, placed before the input projection layer, and **Norm2**, placed after the SSM block. Our methodology is organized in **two stages**: the first stage analyzes stability with LN at Norm2, and the second stage examines optimization with BN at Norm1. For stability, we track the spectral norm of the output projection weights and the singular values of input and output-gradient covariance matrices, which capture activation scaling and gradient dynamics across layers. For optimization, we employ Kronecker-Factored Approximate Curvature (K-FAC) analysis to evaluate eigenvalues and condition numbers, thereby characterizing the conditioning of the optimization landscape. Finally, we integrate these insights into a composite normalization strategy (**BN→SSM→LN**) and systematically evaluate its effectiveness across vision, language, and long-sequence benchmarks.

### 3.1 PRELIMINARIES OF MAMBA PIPELINE

We first briefly review the core components of the Mamba architecture. As illustrated in Figure 4, let $N_1$ denote the first normalization layer applied to input $x$. The main branch then proceeds through a sequence of transformations:

$$f = N_2(\text{SSM}(F_1(N_1(x)))). \tag{1}$$

Here, $F_1$ represents the main forward path, including a linear projection, depthwise separable convolution, and a SiLU activation. SSM denotes the selective structured state space module, and $N_2$ is the second normalization layer applied after SSM. Meanwhile, in the parallel branch, the normalized input $N_1(x)$ is processed by a lightweight path $F_2$:

$$p = F_2(N_1(x)). \tag{2}$$

$F_2$ includes a linear projection and a SiLU activation. The outputs of the two branches are combined element-wise and followed by a linear layer:

$$y = L(f \otimes p). \tag{3}$$

Where $\otimes$ denotes element-wise multiplication.

## 3.2 Two-Stage Normalization Strategy

Building on our training dynamics analysis, we propose a two-stage normalization strategy that explicitly integrates stability and optimization considerations into the Mamba pipeline.

**Stage 1:** To suppress activation amplification and stabilize training, we place **Layer Normalization** after the SSM block. Formally,

$$f = \text{LN}(\text{SSM}(F_1(\text{Norm1}(x)))). \tag{4}$$

**Stage 2:** To improve conditioning and accelerate convergence, we place **Batch Normalization** before the input projection. The parallel branch then becomes

$$f = \text{LN}(\text{SSM}(F_1(\text{BN}(x)))), p = F_2(\text{BN}(x)). \tag{5}$$

This composite strategy can thus be summarized as a BN→SSM→LN pipeline, combining the stabilizing effect of LN with the optimization benefits of BN.

## 3.3 Stability Metrics

To evaluate training stability, we investigate three statistics, the magnitude of layer input (indicated by $\lambda_{\max}(\Sigma_x)$), the magnitude of layer output-gradient (indicated by $\lambda_{\max}(\Sigma_{\nabla h})$) and the magnitude of output projection weight (indicated by $\|W\|_2$) which ensure that weight magnitudes can grow under gradient descent while gradient norms shrink proportionally—thereby avoiding divergence. To quantify this, we adopt two spectral metrics at the output layer, as shown in Figure 4.

- The spectral norm of the output projection weights , reflecting the scale of activations during training.
- The maximum eigenvalues of the input activation covariance and output gradient covariance matrices, indicating sensitivity to scale perturbations in forward and backward propagation.

Since SSM amplifies activations in forward propagation and accumulates over depth, **Norm2** is used to regulate the activation scale. Prior work has also shown that LayerNorm stabilizes training. We therefore conduct controlled experiments comparing `None→SSM→None` and `None→SSM→LN`, and analyze the impact of LN on Mamba training stability using the above metrics.

These metrics jointly indicate whether normalization reduces distortion in activations and gradients across layers, thereby stabilizing overall training dynamics. The subsequent experimental results on Mamba further validate our analysis.

## 3.4 Optimization Metrics

To evaluate the impact of **Norm1** on model trainability, we analyze the spectral structure of the Fisher Information Matrix (FIM), which characterizes the curvature of the loss landscape, as shown in Figure 4. However, due to memory and compute constraints, directly analyzing the full curvature matrix is infeasible. We instead approximate it using Kronecker-Factored Approximate Curvature (K-FAC) (Huang et al., 2020; Martens & Grosse, 2015). The FIM can be approximated as a block-diagonal matrix:

$$\mathcal{F} \approx \begin{bmatrix} \mathcal{F}_1 & 0 & \cdots & 0 \\ 0 & \mathcal{F}_2 & \cdots & 0 \\ \vdots & \vdots & \ddots & \vdots \\ 0 & 0 & \cdots & \mathcal{F}_L \end{bmatrix}, \tag{6}$$

The $k$-th block $\mathcal{F}_k$ (or the $k$-th layer) is approximated as:

$$\mathcal{F}_k \approx \mathbb{E}[x_k x_k^\top] \otimes \mathbb{E}\left[\frac{\partial l}{\partial h_k}\left(\frac{\partial l}{\partial h_k}\right)^\top\right], \tag{7}$$

where, $x_k$ is the input to the $k$-th layer, and $\frac{\partial l}{\partial h_k}$ is the output gradient.

To investigate the optimization role of Norm1 in Mamba, we conduct experiments comparing `None→SSM→LN` and `BN→SSM→LN`. We analyze the impact of BN using the condition number $\kappa(\mathcal{F}_k)$[1]:

$$\kappa(\mathcal{F}_k) = \frac{\lambda_{\max}(\mathcal{F}_k)}{\lambda_{\min}(\mathcal{F}_k)} \tag{8}$$

A lower $\kappa(\mathcal{F}_k)$ implies better conditioning and more efficient gradient-based optimization. A higher condition number indicates ill-conditioning and potential convergence challenges. The experimental results on the Mamba architecture presented later also support this analysis.

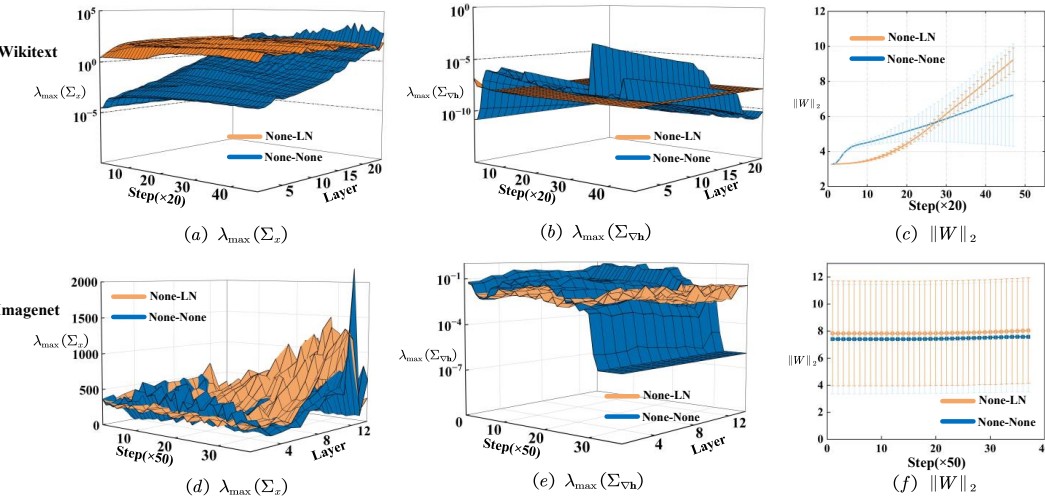

(a) $\lambda_{\max}(\Sigma_x)$      (b) $\lambda_{\max}(\Sigma_{\nabla h})$      (c) $\|W\|_2$

(d) $\lambda_{\max}(\Sigma_x)$      (e) $\lambda_{\max}(\Sigma_{\nabla h})$      (f) $\|W\|_2$

Figure 5: Analysis of layer input magnitude, output gradient magnitude, and weight norm. Yellow indicates `None→LN`, and blue indicates `None→None`. Subfigures (a), (b), and (c) illustrate the variations in stability metrics on the WikiText-103 dataset, while (d), (e), and (f) present the corresponding results on the ImageNet-100 dataset.

In the following section, we conduct experiments across diverse tasks to validate the effectiveness and generalizability of the proposed method.

## 4 EXPERIMENTS

In this section, we first introduce the datasets and experimental settings used to evaluate the impact of normalization on the Mamba architecture across vision, natural language processing, and sequential tasks. Next, we analyze the normalization results in language modeling and image classification tasks using output-layer weight norms, eigenvalues of input-gradient covariance matrices, and K-FAC condition numbers. Finally, we conduct comparison experiments on our proposed composite BN and LN normalization strategy across various tasks to verify its generalizability.

---

[1]The general condition number with respect to the percentage is defined as: $\kappa_p = \frac{\lambda_{\max}}{\lambda_p}$ where $\lambda_p$ s the $p - th$ eigenvalue (in descending order). This measure provides a better characterization of over-parameterized models.

## 4.1 EXPERIMENT SETTINGS

**Baselines**   We select the vanilla Mamba architecture, which adopts RMSNorm-None as the normalization configuration, and the widely used VMamba architecture, which employs LN-LN as its normalization setup. Both serve as baselines for comparison.

**Datasets**   We use a range of datasets to evaluate the performance across different tasks. For stability analysis and optimization analysis, we utilize WikiText-103 (Merity et al., 2016), a widely-used dataset for language modeling, and ImageNet-100 (ima, 2019), a subset of ImageNet for image classification. For generalization verification, we evaluate the combined normalization strategy across a variety of benchmark datasets, including sequence tasks from the LRA Benchmark (Tay et al., 2021), NLP tasks with WikiText-103, and computer vision tasks such as ImageNet-100, COCO (Lin et al., 2014) and ADE-20K (Zhou et al., 2019). The dataset and experimental configurations are described in detail in the Appendix 1.

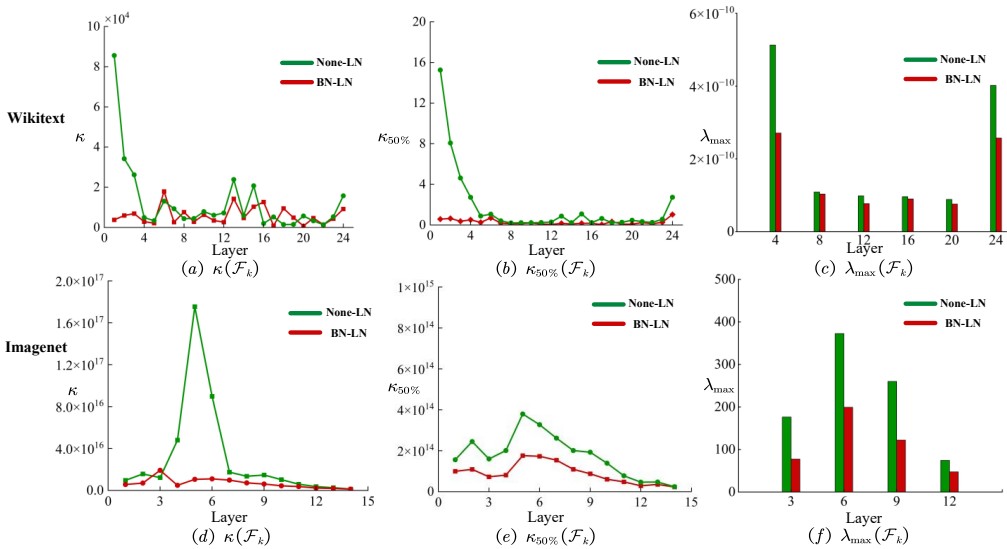

Figure 6: Analysis of the condition of K-FAC (indicated by $\kappa$ and $\kappa_{50\%}$) and magnitude of K-FAC (indicated by $\lambda_{\max}(\mathcal{F}_k)$). The green line represents None→LN, and red line indicates BN→LN. Subfigures (a), (b), and (c) illustrate the corresponding optimization metric changes on the WikiText-103 dataset, while (d), (e), and (f) are on the ImageNet-100 dataset.

## 4.2 STABILITY ANALYSIS

We begin by examining the impact of normalization strategies on training stability. Following the setup described in the Method section, we compare two configurations: None→SSM→LN and None→SSM→None, which correspond to applying LayerNorm after the SSM versus no normalization at all.

On the WikiText-103 and ImageNet-100 dataset, we track the spectral norm of output projection weights, as well as the maximum eigenvalues of the input activation covariance and output gradient covariance matrices across Mamba layers, as shown in Figure 5. The results are summarized below:

- **Weight Norms**: Under the None→SSM→None configuration, the weight norms of deeper layers (e.g., layer 20) increase significantly, far exceeding earlier layers. This results in gradient explosion and even training divergence. In contrast, with LayerNorm (None→SSM→LN), the norm trends remain consistent across layers, and gradients maintain scale invariance, enabling smoother training, as shown in Figures 5(a) and (d).

- **Output Gradient Eigenvalues**: Compared to the None→SSM→None, None→SSM→LN exhibits more consistent gradient eigenvalue distributions and reduced fluctuations during training, suggesting smoother gradient flow, as shown in Figures 5(b) and (e).

- **Input Covariance Eigenvalues**: The None→SSM→LN setup maintains consistent and relatively high eigenvalues across layers, with minimal variation over training iterations, indicating effective suppression of forward-pass scale perturbation. Without normalization, inter-layer eigenvalue differences are large, reducing numerical stability, as shown in Figures 5(c) and (f).

These results confirm that the normalization after SSM (**Norm2**) significantly suppresses activation and gradient explosion, thereby improving the training stability of deep Mamba networks. This also validates the theoretical insights in Section 3, where Norm2 was shown to alleviate scale inflation caused by the SSM.

### 4.3 OPTIMIZATION ANALYSIS

We further investigate the effect of input-side normalization (**Norm1**) on optimization efficiency by applying BN before the SSM. We compare BN→SSM→LN against None→SSM→LN, using the maximum eigenvalue and condition number of the K-FAC-approximated Fisher Information Matrix as evaluation metrics.

Results on the WikiText-103 and ImageNet-100 datasets show that:

- **K-FAC Condition Number**: Across 100% and 50% thresholds, the condition numbers under BN→SSM→LN ( with BN) are significantly lower than those without BN (None→SSM→LN), indicating faster gradient convergence and improved training efficiency, as shown in Figures 6(a) and (d).
- **Convergence Performance**: Compared to the None→SSM→LN (without BN), BN→SSM→LN (with BN) helps the Mamba reach lower training loss and better generalization performance more rapidly, as shown in Figures 6(b) and (e).
- **K-FAC Maximum Eigenvalue**: The K-FAC Maximum Eigenvalue under BN→SSM→LN are lower than under None→SSM→LN, suggesting better alignment in parameter update directions and a smoother optimization landscape, as shown in Figures 6(c) and (f).

These results indicate that input-side BN not only accelerates convergence but also improves numerical conditioning during optimization, thereby enhancing the trainability of Mamba models.

### 4.4 VALIDATION OF BN-LN COMPOSITE NORMALIZATION

Building on the above theoretical and empirical analyses, we propose the composite normalization strategy BN→SSM→LN, and conduct systematic comparisons across tasks including vision classification, segmentation, and reasoning, sequence modeling, natural language processing,. The datasets include ImageNet-100, COCO, ADE-20K, Pathfinder, ListOps, CIFAR-10, IMDB (Text), and WikiText-103. Results are summarized in Tables 1, 2, and 3, respectively.

Table 1: Results of different normalization strategies on sequence tasks. Configurations that result in divergent (NaN) losses during training are marked with an asterisk (*).

| Method | ListOps | CIFAR | Pathfinder |
|---|---|---|---|
| None→None | 38.61* | 56.4* | 49.95 |
| RMSN→None | 39.51 | 62.74 | 51.00 |
| BN→BN | 37.50* | 63.09 | 50.80* |
| LN→LN | 42.18 | 58.80 | 50.80 |
| **BN→LN (Ours)** | **43.75** | **63.41** | **51.43** |

We can observe that single-use BN or LN strategies lead to unstable or divergent behavior in certain tasks. In contrast, the BN-LN composite strategy not only significantly accelerates convergence but also achieves the best (or even state-of-the-art) performance across all evaluated tasks. Particularly in deeper Mamba models, BN-LN effectively balances optimization speed and training stability, demonstrating stronger generalization.

Table 2: Results of different normalization strategies on NLP task WikiText-103. Configurations that result in divergent (NaN) losses during training are marked with an asterisk (*).

| Method | WikiText-103 | IMDB |
|---|---|---|
| None→None | 201.07* | 77.2* |
| RMSN→None | 28.9 | 78.40 |
| BN→BN | 201.3* | 70.24 |
| LN→LN | 27.59 | 79.87 |
| **BN→LN (Ours)** | **27.57** | **81.48** |

Table 3: Results of different normalization strategies on visual tasks.Configurations that result in divergent (NaN) losses during training are marked with an asterisk (*).

| Method | ImageNet100 | COCO | ADE20K |
|---|---|---|---|
| None→None | 11.52* | 0* | 0* |
| RMSN→None | 87.04 | 24.2* | 26.17 |
| BN→BN | 44.92* | 20.1 | 25.78 |
| LN→LN | 87.04 | 34.5 | 26.92 |
| **BN→LN (Ours)** | **87.74** | **34.9** | **27.32** |

Moreover, the evaluation metrics curves during training are shown in 7. These figures also demonstrate that combined normalization leads to faster convergence. For example, in the segmentation task, the combined normalization consistently outperforms single normalization methods in terms of accuracy and reaches the highest accuracy earlier during training.

To further evaluate the effectiveness of different normalization strategies in accelerating convergence, we replaced **Norm1** with several commonly used normalization methods, as shown in Figure 8. It can be seen that the combined normalization configuration of BN+LN not only maintains high final accuracy but also achieves the fastest convergence, making it the optimal choice for efficient training.

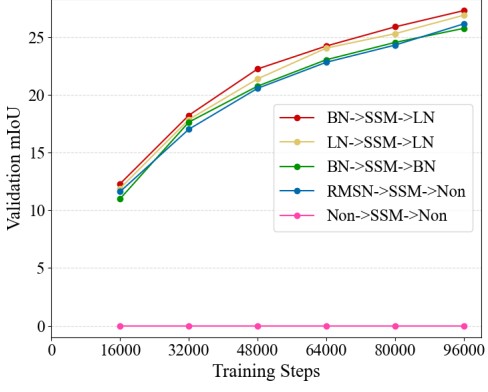 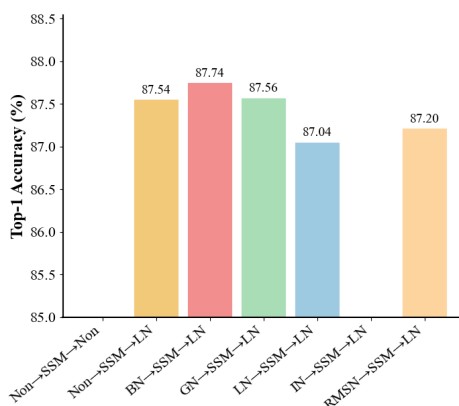

Figure 7: Training stability and convergence on ADE-20K segmentation.

Figure 8: Comparison with various Norm1 methods.

## 5 CONCLUSION

In this paper, we investigate the training stability and optimization convergence of normalization in the Mamba architecture. For training stability, we monitor the spectral norm of the output projection weights and the maximum eigenvalue of the output layer covariance. These analyses show that

post-SSM LayerNorm is essential for suppressing activation and gradient amplification, thereby preventing gradient explosion in deep networks. For optimization efficiency, through condition-number estimates of a K-FAC-approximated Fisher Information Matrix, we show that pre-SSM BatchNorm substantially improves numerical conditioning, accelerating gradient convergence and training speed. Finally, across tasks such as language modeling, image classification, and semantic segmentation, our composite strategy not only converges more rapidly but also outperforms baselines. However, this study is limited by its focus on only two normalization methods and fixed insertion positions, which may restrict scalability to deeper networks and larger-scale tasks. Future work will explore broader normalization variants, automated placement strategies, and extend the framework to more complex architectures and large-scale settings to enhance generality and performance.

## ETHICS STATEMENT

This research strictly adheres to the ICLR Code of Ethics. Our work does not involve human subjects, nor does it use datasets containing sensitive, private, or discriminatory content. All datasets employed are publicly available and used in compliance with their licenses and release practices. The methods and conclusions of this research do not contain potential malicious applications, and we have carefully evaluated and avoided possible negative societal impacts. There are no conflicts of interest or inappropriate sponsorship involved, and all experiments and results comply with research integrity and academic standards.

## REPRODUCIBILITY STATEMENT

We have made every effort to ensure the reproducibility of our results. The paper and appendix include detailed descriptions of the model architecture, algorithmic procedures, and experimental settings. All datasets used are publicly available, and the preprocessing steps are documented in the supplementary materials. We will release our complete source code, training scripts, and experimental configurations, allowing other researchers to independently reproduce our experiments and main findings.

## LLM USAGE STATEMENT

A large language model (LLM) was employed solely for grammatical error checking during the preparation of this manuscript. The LLM was not used for generating research ideas, designing experiments, analyzing results, or writing substantive scientific content. All methodological and experimental contributions are the authors' own work.

## IMPACT STATEMENT

This paper presents work whose goal is to advance the field of Deep Learning. There are many potential social consequences of our work, none which feel must be specifically highlighted here.

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

# A  APPENDIX

In this appendix, we provide additional details that could not be included in the main paper due to limited space, which comprises the details of our experiment settings, theoretical backgrounds, empirical experiments, and theoretical analysis. We discuss:

- Datasets and experiment settings.
- Supplementary Theoretical Backgrounds
- Empirical evidence for the role of **Norm2** in enhancing **training stability**.
- Empirical evidence for the role of **Norm1** in improving **optimization condition**.
- Discussion of other composite Strategy cases.

## A  DATASETS AND EXPERIMENT SETTINGS

We conduct experiments on benchmark datasets spanning three domains: sequential modeling, natural language processing (NLP), and computer vision (CV).

### A.1  NLP TASK

**Dataset**  WikiText-103 (Merity et al., 2016) is a large-scale English word-level language modeling benchmark consisting of 28,475 high-quality Wikipedia articles. It retains original casing, punctuation, and numerical content, with the training set comprising approximately 103 million words and a vocabulary of over 260,000 tokens. The validation and test sets each contain 60 full articles. Notably, the dataset preserves paragraph continuity within articles, making it well-suited for evaluating a model's ability to capture long-range dependencies across thousands of tokens.

**Experiment Setting**  Our Mamba-based language model comprises 24 layers with a hidden dimension of 768, totaling approximately 125 million parameters. The model adopts the Mamba1 state space architecture, without employing RMS normalization or tying input and output embeddings. We trained the 24-layer Mamba for 150 epochs using Distributed Data Parallel (DDP) across 8 GPUs, with a global batch size of 128. We use the AdamW optimizer (Loshchilov & Hutter, 2017) with a peak learning rate of $1.5 \times 10^{-3}$ and a weight decay of 0.25. The learning rate follows a cosine annealing schedule with 1% linear warm-up steps, starting from $1 \times 10^{-6}$ and decaying to 10% of the peak value. Gradient clipping is applied with a maximum norm of 1.0. All computations are performed using FP32 precision.

### A.2  SEQUENTIAL MODELING BENCHMARK

In Long range arena(LRA) benchmark (Tay et al., 2021), we use a 8-layer Mamba1-based sequence classification model with a hidden size of 128 and approximately 1.4M parameters. It uses a state dimension of 64, kernel size of 4, expansion factor of 2, and no normalization layers. Positional encodings are added to capture sequence order.

**Dataset**  ListOps (Nangia & Bowman, 2018) contains approximately 90,000 training samples, 10,000 validation samples, and 10,000 test samples, totaling around 110,000 prefix-style arithmetic expressions with nested operations. Each sequence has an average length of 130 tokens, with some exceeding 200 tokens. The task requires outputting a single integer between 0–9, with operators such as MAX, MIN, MED, and SUM_MOD (SM). This dataset evaluates a model's ability to reason over long-distance dependencies and recursive tree structures, using accuracy as the evaluation metric.

**Experiment Setting**  The model is trained on LISTOPS for 40 epochs using AdamW (learning rate $1 \times 10^{-4}$, weight decay 0.05) with a constant schedule and 2,000 warm-up steps. Training is performed with DDP over 8 GPUs, batch size 64, and no gradient clipping. Inputs are padded to 2,048 tokens with a vocabulary size of 18. End-of-sequence tokens are appended, and outputs are aggregated via pooling with length-aware processing.

**Dataset** IMDB (Maas et al., 2011) is a sentiment analysis dataset consisting of English movie reviews. It provides 25,000 labeled training samples, 25,000 labeled test samples, and an additional 50,000 unlabeled reviews for semi-supervised learning. Review lengths range from 200 to 1,000 words, and labels are binary: positive or negative. The dataset features a balanced sentiment distribution and linguistic diversity, including slang, negation, and sarcasm, making it a standard benchmark for assessing a model's capacity to capture fine-grained sentiment in long-form text. Accuracy is used as the evaluation metric.

**Experiment Setting** The model is trained on the IMDB dataset for 65 epochs with a batch size of 32. We use the AdamW optimizer with a learning rate of $1 \times 10^{-4}$, a weight decay of 0.1, and a constant learning rate schedule with 2,000 linear warm-up steps (approximately one epoch). Input sequences are tokenized at the character level using a minimum frequency threshold of 15, yielding a vocabulary of 135 characters. Sequences are padded to a maximum length of 4,096 characters, and end-of-sequence tokens are appended. The final outputs are computed via pooling-based sequence classification with length-aware aggregation to effectively handle variable-length movie reviews.

**Dataset** CIFAR-10 (Krizhevsky et al., 2009) consists of 50,000 training images and 10,000 test images, totaling 60,000 32×32 RGB images across 10 common categories: airplane, automobile, bird, cat, deer, dog, frog, horse, ship, and truck. The low resolution and cluttered backgrounds demand that models learn discriminative features from limited pixels. Each class contains 5,000 training samples. The dataset is widely used to assess image classification capabilities, with accuracy serving as the evaluation metric.

**Experiment Setting** The model processes CIFAR-10 images converted to grayscale and serialized into 1,024-token sequences (32×32 pixels), without any data augmentation. Training is conducted for 150 epochs with a batch size of 50. We use the AdamW optimizer with a learning rate of $1 \times 10^{-3}$, weight decay of 0.1, and $\beta$ parameters set to (0.9, 0.95). The learning rate follows a cosine annealing schedule with 2,000 linear warm-up steps. Gradient clipping is applied with a maximum norm of 1.0 to further stabilize training. Each pixel is treated as a discrete token, allowing sequence modeling techniques to be applied to vision tasks through this serialization-based approach.

**Dataset** Pathfinder (Linsley et al., 2018) is a visual reasoning benchmark designed to assess topological reasoning. The task is to determine whether two marked circles in a binary image are connected by a single continuous path. The dataset includes approximately 100,000 training images and 20,000 validation/test images, with each image sized at 64×64 pixels. As the number of path segments increases, so does task difficulty. Since no semantic cues are available, models must rely on global receptive fields and spatial reasoning. Evaluation is based on accuracy.

**Experiment Setting** The model processes PATHFINDER images converted to grayscale and serialized into sequences of varying lengths: 1,024 tokens (32×32), 4,096 tokens (64×64), and 65,536 tokens (256×256). Training is conducted for 200 epochs with a batch size of 32. We employ the AdamW optimizer with learning rates of $1 \times 10^{-4}$ for lower resolutions and $1 \times 10^{-3}$ for the 256×256 setting, using weight decay values between 0.05 and 0.1, and $\beta$ parameters set to (0.9, 0.95). The learning rate follows either a constant or cosine annealing schedule with 5,000 linear warm-up steps. Gradient clipping with a maximum norm of 1.0 is applied for higher-resolution inputs. Each pixel is treated as a discrete token in the serialized sequence, allowing us to evaluate the model's capacity to capture long-range dependencies across different input lengths.

A.3 COMPUTER VISION(CV) TASK

We conduct experiments on the ImageNet-100, COCO2017, and ADE20K datasets using the open-source `Vanilla-VMamba-Tiny` model. To ensure a fair comparison, we retrain each configuration from scratch on an 8-GPU server without employing any pre-trained weights. This avoids inconsistencies caused by potential mismatches between modified normalization layers and pre-trained parameters. Furthermore, for each dataset, we adopt the same hyperparameter settings as in the original implementation. The details are as follows:

**Dataset** ImageNet-100 (ima, 2019) is a curated subset of ImageNet-1k (Deng et al., 2009) , comprising 100 randomly selected and semantically coherent classes. Each class contains 1,300 training images and 50 validation images, totaling 130,000 training and 5,000 validation samples. Image resolution follows that of the original ImageNet, commonly resized by cropping or scaling the short edge to 160–224 pixels. It is used to evaluate image classification performance, covering common entities such as animals, objects, and scenes, with Top-1 accuracy as the evaluation metric.

**Experiment Setting** The backbone consists of 14 Mamba blocks with three downsampling stages, and the layer configuration is set to [2, 2, 8, 2]. We train the model using the AdamW optimizer with a weight decay of 0.05, an initial learning rate of $5 \times 10^{-3}$, a batch size of 256, and a total of 300 epochs.

**Dataset** COCO 2017 (Lin et al., 2014) is one of the most widely used benchmarks for multi-task vision evaluation, featuring approximately 330,000 images, including 118,000 for training, 5,000 for validation, and 20,000 for test-dev. It includes 80 object detection categories and 91 stuff categories, with 1.5 million object instances annotated with bounding boxes. The images depict real-world scenarios with dense object layouts, occlusions, and large scale variations. It serves as a standard benchmark for object detection and instance segmentation, with mean Average Precision (mAP) used for evaluation.

**Experiment Setting** The backbone consists of 14 Mamba blocks with three downsampling stages, and the layer configuration is [2, 2, 9, 2]. For object detection and instance segmentation, we employ the Mask R-CNN head. The training is performed using the AdamW optimizer with a weight decay of 0.05, an initial learning rate of $1 \times 10^{-4}$, a batch size of 8, and a total of 12 epochs.

**Dataset** ADE-20K (Zhou et al., 2019) is a benchmark for semantic segmentation and scene parsing, aggregating over 27,000 scene images from the SUN and Places datasets. All images are annotated with pixel-level polygons, covering 150 semantic classes and over 3,000 instance-level object categories. The dataset spans a wide variety of environments, including indoor, outdoor, natural, and urban scenes, with annotations for both visible and occluded object regions. It is the standard evaluation set for fine-grained segmentation and multi-scale understanding, using mean Intersection over Union (mIoU) as the evaluation metric.

**Experiment Setting** The backbone consists of 14 Mamba blocks with three downsampling stages, and the layer configuration is [2, 2, 8, 2]. For semantic segmentation, we use the UPerHead as the decoding head. Training is conducted using the AdamW optimizer with a weight decay of 0.01, an initial learning rate of $6 \times 10^{-5}$, a batch size of 32, and a total of 160,000 training iterations.

## A.4 SUPPLEMENTARY THEORETICAL BACKGROUNDS

In our section Method, we presented the definitions of the stability and optimization metrics along with the associated empirical conclusions. In this appendix, we provide supplementary theoretical discussion to further substantiate the principles underlying our analysis of training stability and optimization behavior in neural networks.

## A.5 STABILITY ANALYSIS

In gradient-based neural network training, instability often manifests as exploding gradients, where gradients grow excessively and lead to numerical failures. Intuitively, this phenomenon typically arises from two sources: the explosion of hidden activations (e.g., large spectral values in forward inputs or backward gradients causing NaNs in the loss), or unbounded growth in network weights due to overly large updates during backpropagation. Accordingly, our stability analysis considers both activation dynamics and weight behavior.

Due to the cumulative feature of transformations across multiple layers, the *out_put* layers in deep neural networks are particularly susceptible to numerical instabilities. Moreover, we observed a similar phenomenon in Mamba, as illustrated in Figure 10. The figure compares the activation magnitude range of the SSM outputs before and after applying normalization. It can be seen that the

State Space Model (SSM) tends to amplify activation magnitudes during the forward pass, and such amplification accumulates progressively in deep networks, eventually leading to gradient explosion. Notably, in computer vision (CV) tasks, the amplification of the input $\mathbf{x}$ by the SSM module in the absence of normalization at the `Norm2` position is significantly greater than in sequential data tasks. As a result, CV models are more susceptible to gradient explosion, as shown in Figure11. Chiang et al. (2024) similarly observed this property of Mamba. Introducing normalization effectively mitigates this accumulation and constrains the numerical range. Therefore, we focus our analysis on the spectral properties of the forward inputs, backward gradients, and output-layer weight matrices of *out_proj* layer in each Mamba block.

Previous studies have shown that spectral analysis of activations and monitoring the norms of weight matrices are effective methods for evaluating training stability in deep neural networks(DNNs) Huang et al. (2020). Motivated by this, we adopt two representative indicators to analyze the stability of the Mamba architecture: (1) the maximum eigenvalue of the activation covariance matrix, which captures the scale and distributional dynamics of the hidden activations, and (2) the spectral norm of the output-layer weight matrices, which reflects the model's scaling behavior and its tendency to produce unstable updates.

We denote the covariance matrix of the layer input as $\Sigma_{\mathbf{x}}^l = \mathbb{E}_{p(x)q(y|x)}\left[\mathbf{x}^{l-1}\left(\mathbf{x}^{l-1}\right)^T\right]$ and the covariance matrix of the layer output-gradient as $\Sigma_{\nabla\mathbf{h}}^l = \mathbb{E}_{q(\mathbf{y}|\mathbf{x})}\left[\frac{\partial\ell^T}{\partial\mathbf{h}^l}\frac{\partial\ell}{\partial\mathbf{h}^l}\right]$, where l is the l-th mamba layer.

- **Maximum eigenvalue of the input covariance matrix:**

$$\lambda_{\max}(\Sigma_{\mathbf{x}}^l) = \max\left\{\lambda \in \mathrm{Spec}(\Sigma_{\mathbf{x}}^l)\right\},$$
$$\Sigma_{\mathbf{x}}^l = \mathbb{E}_{p(x)q(y|x)}\left[\mathbf{x}^{l-1}(\mathbf{x}^{l-1})^\top\right], \quad (9)$$

  which measures the second-order statistics of the forward input to the layer.

- **Maximum eigenvalue of the output-gradient covariance matrix:**

$$\lambda_{\max}(\Sigma_{\nabla\mathbf{h}}^l) = \max\left\{\lambda \in \mathrm{Spec}(\Sigma_{\nabla\mathbf{h}}^l)\right\},$$
$$\Sigma_{\nabla\mathbf{h}}^l = \mathbb{E}_{q(y|x)}\left[\frac{\partial\ell^\top}{\partial\mathbf{h}^l}\frac{\partial\ell}{\partial\mathbf{h}^l}\right], \quad (10)$$

  which reflects the second-order sensitivity of the loss with respect to the layer outputs.

A.6 Optimization Analysis

In prior work, the condition number has been widely regarded as a key indicator for monitoring the optimization behavior of deep neural networks Saarinen et al. (1993); Desjardins et al. (2015); Huang et al. (2020). It is formally defined as:

$$\kappa(\mathcal{F}_k) = \frac{\lambda_{\max}(\mathcal{F}_k)}{\lambda_{\min}(\mathcal{F}_k)}, \quad (11)$$

where $\mathcal{F}_k$ denotes the Fisher Information Matrix (FIM) or its approximation for the $k$-th layer.

Intuitively, a condition number closer to 1 implies that the optimization landscape is more isotropic (i.e., closer to a spherical shape). This indicates that weight updates are more evenly distributed across the principal directions of the data, thereby facilitating more stable and efficient convergence.

FIM characterizes the curvature of the loss landscape very well (Oczkowski & Barreca, 1997; Fasina et al., 2023). One successful example is approximating the Fisher Information Matrix (FIM) of DNNs using the Kronecker-factored Approximate Curvature (K-FAC) method Martens & Grosse (2015). In the K-FAC approach, two assumptions are made: (1) weight gradients in different layers are assumed to be uncorrelated; (2) the input and output gradients in each layer are approximated as independent.

Under these assumptions, the full FIM can be approximated as a block diagonal matrix:

$$\mathbf{F} \approx \mathrm{diag}(F^1, F^2, \ldots, F^L),$$

where $F^l$ is the sub-FIM corresponding to the parameters in the $l$-th layer, computed as:

$$F^l = \mathbb{E}_{p(\mathbf{x}), q(\mathbf{y}|\mathbf{x})} \left( \left( \mathbf{x}^{l-1} \left( \mathbf{x}^{l-1} \right)^T \right) \otimes \left( \frac{\partial \ell^T}{\partial \mathbf{h}^l} \frac{\partial \ell}{\partial \mathbf{h}^l} \right) \right)$$

$$\approx \mathbb{E}_{\mathbf{x} \sim p(\mathbf{x})} \left[ \mathbf{x}^{l-1} \left( \mathbf{x}^{l-1} \right)^T \right] \otimes \mathbb{E}_{(\mathbf{x},\mathbf{y}) \sim p(\mathbf{x}) q(\mathbf{y}|\mathbf{x})} \left[ \frac{\partial \ell^T}{\partial \mathbf{h}^l} \frac{\partial \ell}{\partial \mathbf{h}^l} \right]$$

**Properties of the Kronecker-structured FIM.** Given the Kronecker-factored approximation of the Fisher Information Matrix (FIM) for the $l$-th layer as

$$F^l = A \otimes B,$$

where $A$ and $B$ are symmetric positive semi-definite matrices representing the input and output-gradient statistics respectively, the spectral properties of $F^l$ satisfy the following:

- **Eigenvalues.** The eigenvalues of $F^l$ are the pairwise products of the eigenvalues of $A$ and $B$:
$$\lambda(F^l) = \{ \lambda_i(A) \cdot \lambda_j(B) \mid \lambda_i \in \lambda(A), \lambda_j \in \lambda(B) \}.$$
In particular, the maximum eigenvalue satisfies:
$$\lambda_{\max}(F^l) = \lambda_{\max}(A) \cdot \lambda_{\max}(B).$$

- **Condition Number.** The condition number of $F^l$ equals the product of the condition numbers of $A$ and $B$:
$$\kappa(F^l) = \kappa(A \otimes B) = \kappa(A) \cdot \kappa(B).$$

# B EMPIRICAL EVIDENCE FOR THE ROLE OF **NORM2** IN ENHANCING TRAINING STABILITY

Previous studies have shown that Layer Normalization (LN) effectively stabilizes model convergence Ba (2016); Xu et al. (2019); Zhang & Sennrich (2019). In particular, research within Transformer architectures has demonstrated that applying LN before the attention module can yield superior performance Kim et al. (2025); Xiong et al. (2020); Shleifer et al. (2022). This raises the question: does a similar principle hold for the Mamba architecture? Specifically, is LN more effective when placed before the SSM block (Norm1), after the SSM block (Norm2), or on both sides?

To investigate this, we conducted experiments on the ImageNet-100 dataset with three configurations: applying LN before the SSM module (Norm1), after the SSM module (Norm2), and on both sides (Norm1 and Norm2). All experiments were conducted using the same model architecture and training settings to ensure fairness. The results are presented in Figure 12.

**Key observations from Figure 12:**

1. **Normalization can stabilize model training:** The baseline configuration (Non→SSM→Non) fails to converge, achieving only 23.08% Top-1 accuracy. This indicates that without normalization, the amplification of activations and gradient explosion in the SSM severely disrupt training. In contrast, all configurations with normalization successfully converge and achieve Top-1 accuracy above 86%.

2. **Post-SSM LN (Norm2) yields the best performance:** When LN is applied only after the SSM (Non→SSM→LN), the model achieves the highest Top-1 accuracy of **86.36%** and exhibits the steepest convergence curve. This suggests that Norm2 directly suppresses the activation norm explosion from the SSM, significantly improving training stability and convergence speed.

3. **Pre-SSM LN (Norm1) is slightly less effective:** Applying LN only before the SSM (LN→SSM→Non) results in a slightly lower peak accuracy of **86.04%** and slower convergence. This implies that Norm1 helps improve the numerical condition of the input features but is less effective than post-normalization in stabilizing the network.

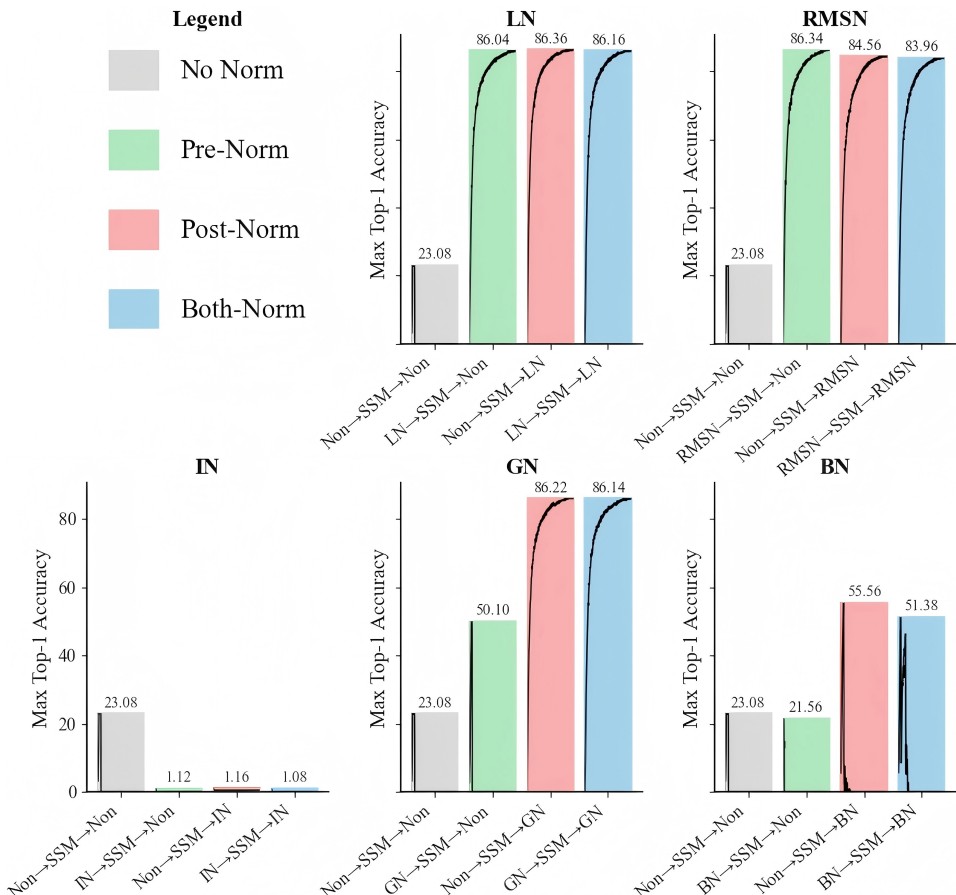

Figure 9: Stability comparison across different normalization methods applied to the SSM block.

4. **Dual LN provides stable convergence but no additional benefit:** Using LN on both sides of the SSM (LN→SSM→LN) yields a final accuracy of **86.16%**, which is between the performances of Norm1 and Norm2 alone. The convergence speed is also intermediate. This suggests that dual normalization does not provide additive benefits and may slightly impair feature expressiveness due to over-normalization.

In summary, LN is effective in stabilizing training in Mamba-based models. However, unlike Transformer models where pre-normalization is often optimal, **post-normalization (Norm2) after the SSM block achieves better stability and performance in Mamba**. While pre-normalization (Norm1) accelerates convergence, it is slightly less effective, and dual normalization brings no additional gains.

### B.1 COMPARISON WITH OTHER NORMALIZATION METHODS

To further validate the generality of post-normalization effectiveness beyond LN, we also evaluated other commonly used normalization strategies, including RMSNorm and GroupNorm. The experimental results are shown in Figure 9.

**Key observations from Figure 9:**

- **RMSNorm and LayerNorm:** Both normalization methods are capable of stabilizing training regardless of placement (pre or post-SSM). For GroupNorm, however, only post-normalization configurations (Non→SSM→GN and GN→SSM→GN) result in stable convergence with Top-1 accuracy exceeding **86.0%**. The pre-normalization setting (GN→SSM→Non) diverges after several training steps, reaching only **50.1%** accuracy.

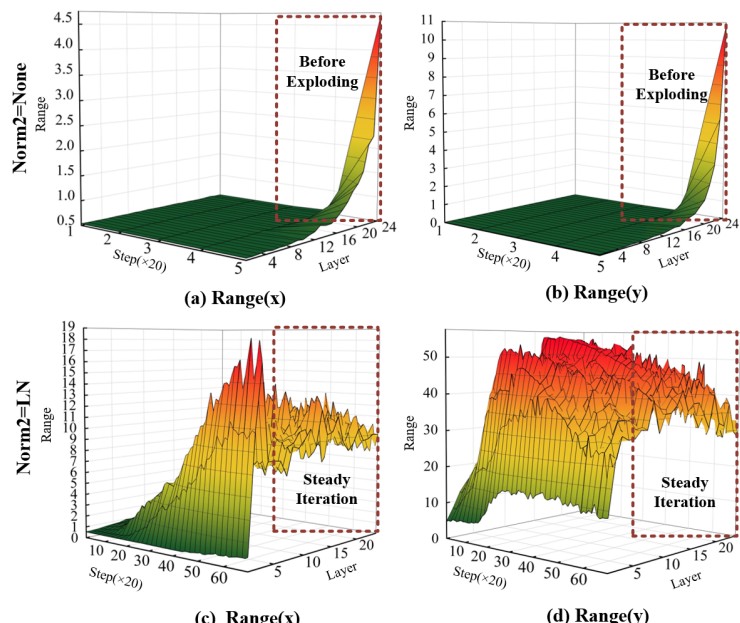

Figure 10: Comparison of the input ($x$) and output ($y$) range of the SSM module on the WIKITEXT-103 dataset. The $x$-axis denotes training steps (logged every 20 steps), and the $y$-axis indicates the layer index of the Mamba block. Subfigures (a) and (b) show the range evolution when no normalization is applied after the SSM layer (`Norm2=None`), while (c) and (d) present the results with LayerNorm applied at Norm2. Applying LN leads to steady iteration dynamics across layers, whereas the absence of normalization results in unstable growth before divergence.

- **InstanceNorm and BatchNorm:** Neither method achieves convergence under any configuration. Post-normalization variants (`Non→SSM→IN` and `Non→SSM→BN`) initially improve performance but subsequently diverge, with Top-1 accuracy peaking at only **1.12%** and **21.56%**, respectively, before collapsing.

These results reinforce the finding that, unlike in Transformer architectures, **post-SSM normalization (Norm2) is particularly effective for ensuring training stability in Mamba models**, especially when using LayerNorm. Accordingly, we adopt `Norm2 = LN` as the default normalization configuration in our main experiments.

## C  EMPIRICAL EVIDENCE FOR THE ROLE OF **NORM1** IN IMPROVING OTIMIZATION CONDITION

### C.1  EFFECT OF BATCH NORMALIZATION ON ACCELERATING CONVERGENCE AND COMPARISON WITH OTHER NORMALIZATION TECHNIQUES

Having established a stable training setup with post-SSM Layer Normalization (Norm2 = LN), we further investigate the effect of Batch Normalization (BN) on accelerating convergence. Specifically, we use the `Non→SSM→LN` configuration as our baseline, and introduce BN before the SSM block (`BN→SSM→LN`, i.e., Norm1 = BN). The experimental results are illustrated in Figure 13.

**Observations from Figure 13:**

With Norm2 fixed as LN to ensure stable training, introducing BN at Norm1 (red line) significantly accelerates convergence compared to the baseline Non+LN (blue line):

- **Faster early-stage convergence:**
  - At epoch 10, `BN+LN` achieves a Top-1 accuracy of approximately 35.6%, compared to 34.8% for `Non+LN`.

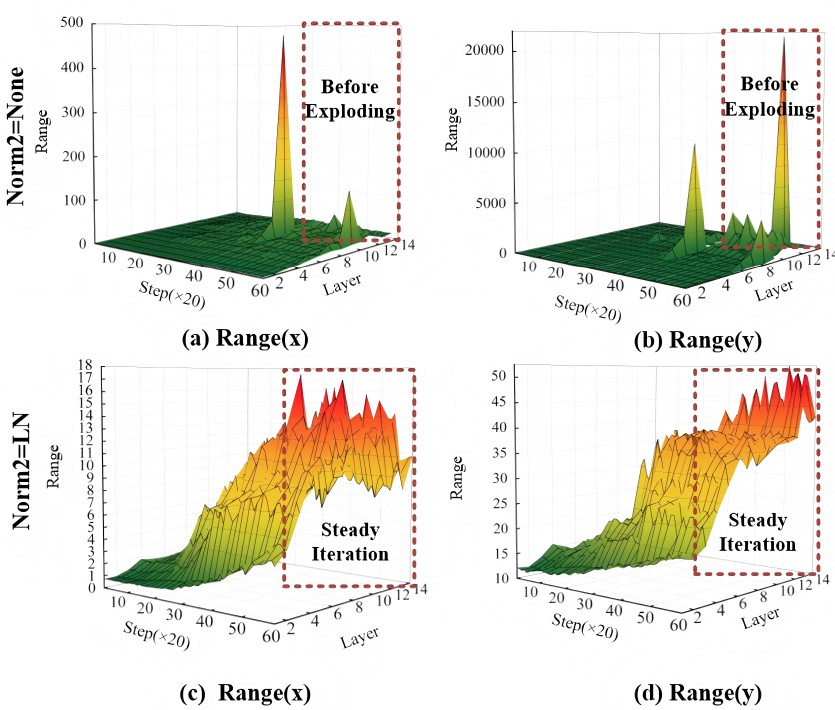

Figure 11: Comparison of the input ($x$) and output ($y$) range of the SSM module on the IMAGENET dataset. The $x$-axis denotes training steps (logged every 20 steps), and the $y$-axis indicates the layer index of the Mamba block. Subfigures (a) and (b) show the range evolution when no normalization is applied after the SSM layer (`Norm2=None`), while (c) and (d) present the results with LayerNorm applied at Norm2. Applying LN leads to steady iteration dynamics across layers, whereas the absence of normalization results in unstable growth before divergence.

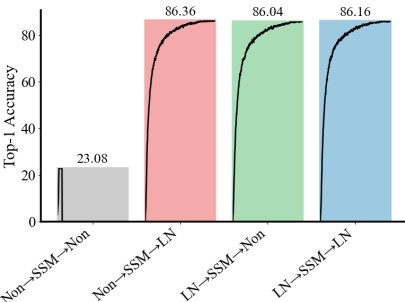

Figure 12: Comparison of training stability with LN applied at different positions in the SSM block. Each bar represents the final Top-1 accuracy, and the inner curve shows the accuracy evolution during training.

- At epoch 20, `BN+LN` reaches about 62.3%, approximately 4 percentage points higher than the baseline.

• **Mid-stage acceleration:**

- From epoch 20 to 50, `BN+LN` maintains a steeper ascent, surpassing 70% accuracy earlier than the baseline.
- By epoch 60, the accuracy curve of `BN+LN` stabilizes around 75%, with reduced oscillation.

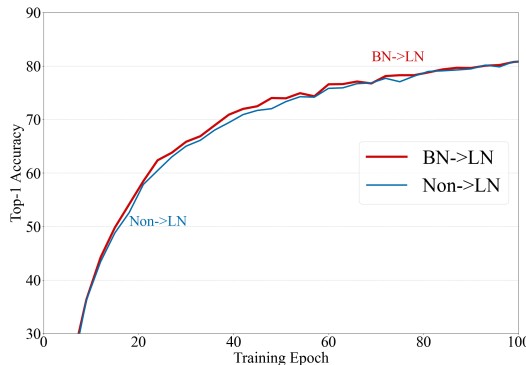

Figure 13: Convergence speed comparison between BN+LN and Non+LN configurations.

These findings suggest that `BN→SSM→LN` enables larger gradient steps during the early training phase, allowing the model to enter the high-accuracy regime faster and resulting in significantly improved convergence speed over `Non→SSM→LN`.

### C.1.1 COMPARISON WITH OTHER NORM1 CONFIGURATIONS

To further assess the acceleration effect across different normalization strategies, we replaced Norm1 with other commonly used normalization methods, including Group Normalization (GN), Root Mean Square Normalization (RMSN), Instance Normalization (IN), and Layer Normalization (LN). We compared their Top-1 accuracy and the steepness of the convergence curves. The results are presented in Figure 14.

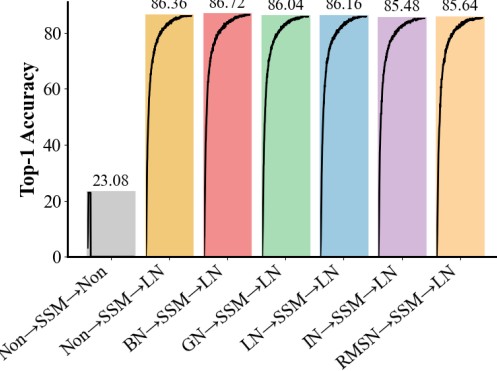

Figure 14: Convergence speed comparison with various Norm1 methods combined with Norm2 = LN.

**Key findings from Figure 14:**

- **BN+LN achieves the best performance:**
  - Final Top-1 accuracy reaches **86.72%**, surpassing the baseline `Non+LN` (86.36%).
  - The learning curve exhibits the steepest slope in the 0–80% range, indicating the most significant early acceleration.
- **Other combinations:**
  - `GN→SSM→LN` and `LN→SSM→LN` also provide faster convergence than the baseline, but with gentler slopes during the first 20–30 epochs.
  - `IN→SSM→LN` and `RMSN→SSM→LN` exhibit the flattest early-stage slopes and the slowest overall convergence, with the lowest final accuracy.

In summary, among all evaluated Norm1 strategies, the **BatchNorm $\rightarrow$ SSM $\rightarrow$ LayerNorm** configuration not only maintains high final accuracy but also maximizes convergence speed, making it the most effective combination for efficient training.

