# OpenReview forum: "Layer-Wise Analysis in Exploring the Normalization Strategies in Mamba"
_ICLR.cc/2026/Conference — ICLR 2026 Conference Withdrawn Submission_

### Official Review · Reviewer_jioT · 2025-10-18

**Soundness:** 1
**Presentation:** 1
**Contribution:** 1
**Rating:** 2
**Confidence:** 4

**Summary:**

This paper investigates how different normalization layer combinations affect the training stability and efficiency of Mamba1 models. The authors identify that the state space module (SSM) tends to amplify activations, leading to instability in deep networks. To address this, they propose a composite normalization strategy (BN→SSM→LN), applying BatchNorm before and LayerNorm after the SSM to balance stability and optimization efficiency. Using Kronecker-Factored Approximate Curvature (K-FAC) analysis, they show that this configuration improves the conditioning of the optimization landscape and accelerates convergence. Experimental results across language, vision, and sequence tasks demonstrate that the BN→SSM→LN scheme consistently outperforms single-normalization or unnormalized baselines. Overall, the study provides a principled normalization guideline for improving the trainability of Mamba-based architectures.

**Strengths:**

The study provides a novel perspective on normalization in state space models (SSMs), addressing an underexplored yet important aspect of SSM training. The use of Kronecker-Factored Approximate Curvature (K-FAC) as a metric to assess model trainability is both innovative and insightful. Furthermore, the proposed normalization configuration is comprehensively evaluated across multiple domains, including natural language processing (WikiText-103, IMDB), computer vision (ImageNet-100, COCO, ADE-20K), and sequential reasoning tasks (ListOps, Pathfinder), demonstrating its robustness and broad applicability.

**Weaknesses:**

While the paper presents an interesting investigation into normalization strategies for SSM-based architectures, I have several concerns:
- Lack of clarity in experimental setup: The authors do not clearly specify the model architectures and their sizes in the main text, which makes it difficult to assess or compare the reported results and accuracies. Although some of these details appear in the appendix, including key configurations and parameter counts in the main paper would improve transparency and reproducibility.
- Insufficient theoretical connection for K-FAC analysis: While the use of Kronecker-Factored Approximate Curvature (K-FAC) as a metric for evaluating trainability is novel, the paper does not establish a clear theoretical link between this metric and the optimization behavior of SSMs, leaving the interpretation somewhat empirical.
- Lack of theoretical justification for the proposed scheme: The BN→SSM→LN normalization strategy, though empirically effective, lacks theoretical grounding or formal analysis to support why this configuration works better than alternatives.

**Questions:**

- Could the authors extend their proposed normalization strategy to the Mamba2 architecture, evaluating its effectiveness across models of different scales (e.g., from 130M to 8B parameters)?
- How does the BN→SSM→LN configuration compare to the default RMSNorm→SSM→RMSNorm normalization used in Mamba2? A direct comparison would help clarify whether the proposed scheme provides consistent benefits across newer Mamba variants.
- What is the rationale for placing normalization before the gated multiplication? In the Mamba2 design (see Figure 6 in the Mamba2 paper), normalization is applied after the gating operation. Could the authors discuss the design difference and its potential implications?
- Could the authors theoretically establish the relationship between K-FAC metrics and model optimization, beyond empirical correlation? This would strengthen the justification for using K-FAC as a measure of trainability.
- Finally, could the authors provide a theoretical explanation or formal proof for why the BN→SSM→LN configuration outperforms other normalization combinations?

---

### Official Review · Reviewer_MVpq · 2025-10-31

**Soundness:** 2
**Presentation:** 2
**Contribution:** 2
**Rating:** 2
**Confidence:** 3

**Summary:**

This paper focuses on addressing the training instability and optimization inefficiency of the Mamba architecture. Authors propose the BN->LN. Next, they use optimization metrics to prof its effectiveness.

**Strengths:**

The author has proposed an excellent research topic. It is indeed true that SSM will lead to the expansion of the activation range, posing relatively significant challenges to training stability. The author has elaborated on the background of the article quite clearly, and the structure of the paper is complete and reasonable.

**Weaknesses:**

Main issues:
1. The main modification of the paper to the Mamba architecture is BN->LN, but I do not figure out a clear motivation for it, i.e., why not other combinations (like LN->BN or other normalization methods)? Though authors provide related ablations, a more theoretical analysis is required.
2. Experimental results are not compared with mentioned baselines in the "Introduction" section.
3. The experiment is not convincing enough, like more models (Mamba-ND) and more taks (zero-shot).

Other issues:
The font size in Figure 4.5.6. is too small.

**Questions:**

Since the author proposes using the method in 3.4 to illustrate the effectiveness of BN->LN, how to demonstrate the effectiveness of the method in 3.4?

---

### Official Review · Reviewer_ti93 · 2025-11-01

**Soundness:** 2
**Presentation:** 3
**Contribution:** 2
**Rating:** 4
**Confidence:** 4

**Summary:**

This paper investigates training instability in the Mamba architecture, focusing on the amplification of activations and gradient explosion as network depth increases. The authors propose a composite normalization strategy (BN→SSM→LN) to improve both stability and optimization efficiency. Through spectral norm, input–output covariance eigenvalue, and K-FAC condition number analyses, they show that post-SSM LayerNorm (LN) stabilizes training, while pre-SSM BatchNorm (BN) accelerates convergence. Experiments across vision, language, and long-sequence benchmarks demonstrate consistent improvements over single normalization or none.

**Strengths:**

1. The paper identifies instability caused by SSM amplification and constructs a logical two-stage normalization solution supported by layer-wise metrics and optimization theory.
2. The use of spectral norms, eigenvalue tracking, and K-FAC condition numbers provides interpretable evidence linking normalization placement to training stability and optimization behavior.

**Weaknesses:**

1. The methodological novelty primarily lies in systematic analysis and positional design rather than new normalization algorithms. The relation to Transformer Pre-LN, DeepNorm, and scaling techniques is discussed but not deeply unified.
2. BatchNorm’s inference-stage instability remains underexplored. Although training improvements are shown, robustness under small-batch or distribution-shift conditions is not evaluated, limiting applicability to large-scale NLP systems[1].
3. The ablation on alternative normalizations (RMSNorm, ScaleNorm, GroupNorm) is limited, leaving unclear the efficiency–accuracy trade-offs and generalization of the findings.
4. The study could benefit from stronger head-to-head comparisons with top-performing Mamba variants such as vim, simba, local mamba, plain mamba, etc.
5. The computation and practicality of real-time K-FAC monitoring for adaptive normalization placement are not evaluated.

[1] Wang et al., *Understanding the Failure of Batch Normalization for Transformers in NLP*, NeurIPS 2022.

**Questions:**

1. How does the BN→SSM→LN configuration balance stability, convergence speed, and throughput compared to LN-only or RMSNorm-only baselines across tasks? Is there a task-adaptive rule or heuristic for normalization placement selection?
2. How are BatchNorm statistics handled during inference to mitigate training–inference inconsistency? Were robust alternatives such as RBN or GhostBN tested under small-batch or domain-shift scenarios?
3. If only one normalization can be used due to efficiency constraints, is there a general rule linking task type, batch size, or data modality to the optimal choice (LN vs BN)?
4. How does the proposed strategy theoretically and empirically relate to Transformer stabilization techniques such as Pre-LN, Post-LN, and DeepNorm [1][2]? Could a unified interpretation framework be provided?

[1] Xiong et al.,*On Layer Normalization in the Transformer Architecture*, 2020.

[2] Shleifer et al.,*Normformer: Improved Transformer Pretraining with Extra Normalization*, ICLR 2022.

---

### Note · Authors · 2026-01-21

I have read and agree with the venue's withdrawal policy on behalf of myself and my co-authors.